# Flow characterization and structural alterations in Ahmed glaucoma FP7 tubes after in-vitro aging in silicone oil

Abu Tahir Taha[1], Matthew Clarke[2], Chiara Wabl[1], Ying Han[1,3], Frank Brodie[1,4]*

1 Department of Ophthalmology, University of California San Francisco, San Francisco, CA, United States of America, 2 ForSight VISION6 Inc, Brisbane, CA, United States of America, 3 The Francis I. Proctor Foundation for Research in Ophthalmology, San Francisco, CA, United States of America, 4 San Francisco Veteran Affairs Medical Center, San Francisco, CA, United States of America

* frank.brodie@ucsf.edu

## Abstract

### Purpose

Patients with intraocular silicone oil (SO) display higher odds of surgical failure after Ahmed glaucoma valve (AGV) implantation compared to patients without SO. However, the structural impact of SO exposure on silicone-made AGV tubes and the resulting changes in flow rate remain unexplored. This in-vitro study evaluated changes in tube dimensions and flow rates of AGV FP7 tubes after SO exposure to inform clinicians how such changes may impact AGV functionality.

### Methods

AGV FP7 tube segments underwent accelerated aging to approximate 90 days of exposure to the following media: Balanced Salt Solution (BSS), 1000 centistokes (cs) SO, and 5000cs SO. Tube dimensions were measured before and after aging. A constant gravity flow test setup was created to measure flow rates through tubes before and after aging. The students' T-test was used to compare the mean change between groups post-aging.

### Results

Post-exposure, 1000cs and 5000cs SO tube segments increased in length by 5.94% and 5.55%, respectively, compared to 0.38% of BSS tubes (P < 0.05 for both). The inner lumen area expanded for tube segments in 1000cs and 5000cs SO by 11.75% and 2.70%, respectively, but contracted for tubes in BSS by -2.70% (P < 0.01 and P = 0.068 for 1000cs and 5000cs SO, respectively). Post aging, the flow rates increased on average by 61.0% and 98.6% for 1000cs and 5000cs SO, respectively, whereas flow rates for BSS tube segments slightly decreased by -4.92%. The difference was statistically significant for BSS vs. SO groups (P < 0.01 for both).

**Data Availability Statement:** All relevant data are within the manuscript and its Supporting Information files.

**Funding:** The author(s) received no specific funding for this work.

**Competing interests:** The authors have declared that no competing interests exist.

## Conclusions

Prolonged exposure to SO structurally altered the AGV FP7 tube segments by expanding their cross-sectional area, potentially leading to increased flow rates. These results may inform clinicians about potential in-vivo interactions in patients with the simultaneous presence of glaucoma drainage devices and intraocular SO.

## Introduction

Glaucoma drainage devices (GDD) remain one of the primary surgical interventions to treat various forms of glaucoma refractory to medical management [1]. Baerveldt glaucoma implant (BGI) and Ahmed Glaucoma valve (AGV) are the two most commonly used GDDs worldwide and use a silicone-based tube [2–4]. Although GDD tube shunts are traditionally inserted into the anterior chamber (AC), alternative locations, such as the ciliary sulcus or vitreous cavity, are being used more frequently for patients with anterior segment pathologies [5, 6].

Posterior segment complications following GDD implantation have been frequently reported in the literature [7–11]. Of those, retinal detachment is reported in up to 5% of cases and carries significant visual morbidity [8, 10, 11]. Patients with previous ocular histories of uveitis, trauma, lattice degeneration, and vitreous syneresis may be predisposed to retinal detachment following GDD surgeries [10]. Silicone oil (SO) is often instilled intraocularly as a tamponading agent to treat retinal tears or detachments [12]. However, prolonged SO endotamponade may also result in SO emulsification, which can migrate beyond the posterior segment into the AC, thus altering the functionality of GDDs [13].

Given the risk of concomitant SO and GDDs, it is important to evaluate their interaction. Previous studies report that eyes with SO endotamponade undergoing Ahmed glaucoma valve (AGV) implantation have a higher risk of surgical failure compared to eyes without SO [14–16]. However, the underlying reason for this phenomenon is unclear. The proposed hypotheses include chronic inflammation secondary to the presence of SO [14, 15], migration of SO in the subconjunctival space through the tube [17], and lastly, formation of a diffusion barrier between the AC and the posterior chamber (PC). In contrast, few studies [7, 10, 18, 19] and case reports [20–22] have described the successes of GDDs, including AGV, in eyes with SO [21].

Although clinical observations are helpful, they do not capture all the physical changes that may occur when two silicone-based products interact for a prolonged period [23]. Therefore, it is valuable to rigorously characterize the nature of the interaction between SO and GDDs. To our knowledge, no published reports discuss SO's functional and structural impact on GDDs, most of which contain a medical-grade silicone rubber tube [24]. Silicone oil, also known as polydimethylsiloxane (PDMS), has a chemical structure very similar to silicone rubber, a cross-linked polymer of siloxanes [25]. Due to the structural similarity between the two compounds, an interaction between PDMS and silicone rubber tubes is expected. Previous studies have reported swelling of PDMS in hydrocarbon solvents, including silicone oil [26, 27], which may contribute to the altered functionality of silicone-based AGVs observed in clinical studies. As such, the gap our study aims to address is to characterize the dimensional changes in silicone-based FP7 AGV tubes and the resultant alterations in the flow dynamics of FP7 tubes once they are exposed to SO. The results of our study may offer potential explanations for the high failure rates of GDDs in eyes with SO instillation.

## Methods

This experimental bench-top study did not require approval by the Human Research Protection Program (HRPP) at the University of California, San Francisco (UCSF) or San Francisco Veterans Affairs Medical Center.

### Accelerated aging

New World Medical Inc. (Rancho Cucamonga, CA) supplied AGVs for this study. We then cut approximately 6mm segments of FP7 tubes and performed accelerated aging while submerged in: balanced salt solution (BSS), SO 1000cs (SILIKON™ 1000, Alcon, Fort Worth TX), or SO 5000cs (ADATO SIL-OL 5000, Bausch & Lomb, Bridgewater, NJ). We used the standard formula for accelerated aging: $days_{chronological} = days_{incubator} \times 2^{\Delta T/10}$, where $\Delta T$ represent the difference between the temperature chosen for aging (65°C) and the reference temperature (37°C) [28–31]. Thus, 13 days of aging at 65°C equates to $13 \times 2^{65-37/10}$, or 90.5 days of chronological aging at physiological temperature, corresponding to the accepted duration of SO endotamponade for retinal detachment (3 months) [12, 32]. To account for any stochastic effects due to accelerated aging, we also added a 'heat control' by aging tubes at 37°C for 13 days in BSS. Each aging condition contained 3 FP7 tubes, which we photographed using a camera (AmScope®, HD205-WU) attached to a microscope before and after aging. We aged all tubes separately in their respective media types in 3mL glass vials. After aging, we rinsed each tube using BSS to remove any adherent oil and dried them up in open air before photographing them and storing them in sealed vials.

### Experimental setup

A constant gravity flow test to assess the flow rate through FP7 tube segments was adapted from Estermann et al. [33, 34]. A schematic of our setup is shown in Fig 1A and 1B. A funnel of 14.5 cm diameter was attached to a 0.5-inch rigid PVC pipe, which was subsequently fitted on a narrower straight coupling socket. The socket's opening was covered with two layers of Parafilm. An opening was created into the parafilm using a 27-gauge cannula. Through that opening, an FP7 tube segment was inserted. The BSS column was filled to a height of 27.4 cm, equivalent to a pressure of 20 mmHg at the top of the inserted FP7 tube segment. This height was determined using the equation $Pressure = \rho gh$ using the density of water for $'\rho'$.

Each trial of the flow test was conducted for a total of 10 minutes. Given the large diameter of the funnel mouth, there was negligible change in the height of the BSS column over the experiment's duration and, correspondingly, an insignificant change in pressure ($< 0.15$ mmHg) at the top of the tube segment. This gravity flow test was performed on all tubes barring heat control tubes (BSS at 37°C) before and after simulated aging at 65°C. Three trials per tube were performed, and the results were averaged. The flow rate was calculated by converting the mass of collected BSS into volume using the density of water and subsequently dividing that volume by the total amount of time.

### Image processing and statistical analysis

Long axis and en-face images of the tubes before and after aging were obtained. ImageJ (National Institutes of Health, Bethesda, Maryland) software was used to measure the length as well as the inner and outer lumen area using an existing scale (Fig 2) [35]. Each tube's percentage change in parameters after aging was calculated across each aging media type. Two-tailed Student's T-tests were used to compare mean percentage change across different aging conditions.

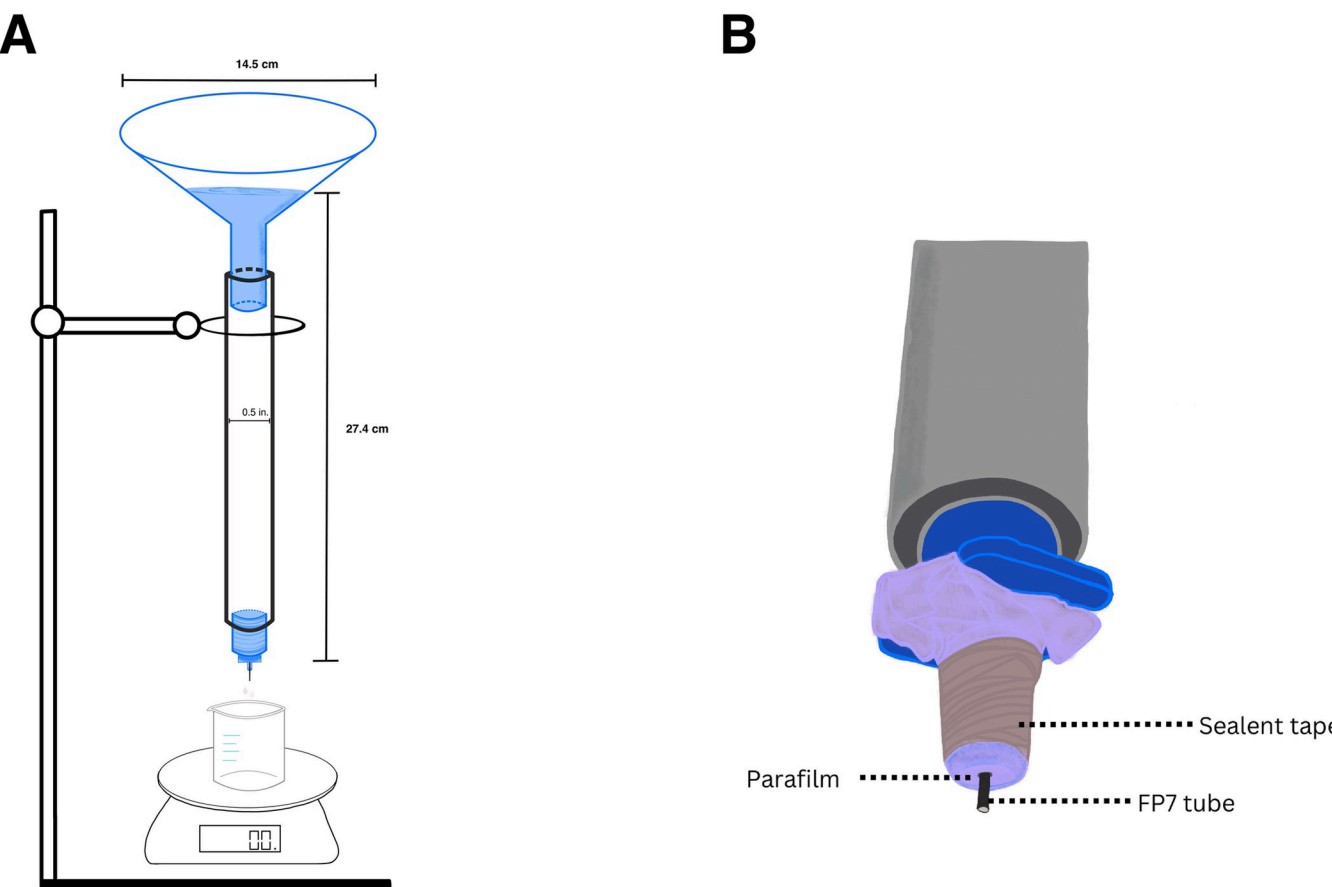

**Fig 1.** (A): Experimental setup for a constant gravity flow test. (B): A zoomed-in version of the setup showing the placement of the FP7 tube.

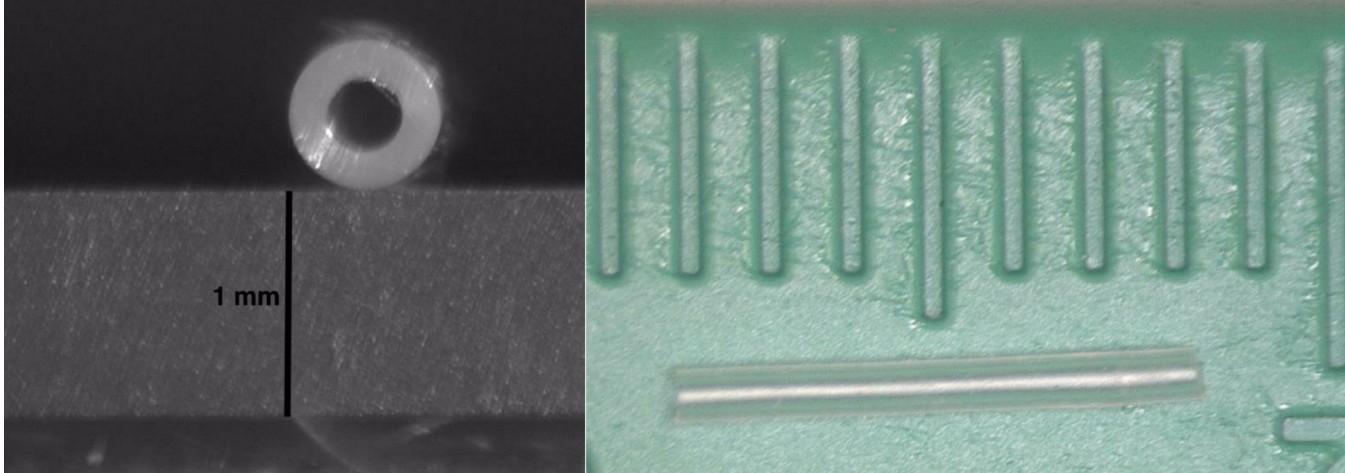

**Fig 2.** A cross-sectional (left) and a longitudinal (right) view of an FP7 tube segment used in this study. The marks on the ruler in the longitudinal view represent an increment of 1cm.

## Results

### Dimensional changes in FP7 tubes after aging

A total of 12 tube segments were included in the study. After simulated aging, none of the tubes displayed any structural deformities. The mean percent change ± standard deviation (SD) in the length of tubes in control (no media), BSS, 1000cs SO and 5000cs SO was -0.42 ± 1.1%, 0.38 ± 1.6%, 5.94 ± 2.0% and 5.55 ± 1.1% respectively. Tubes aged in BSS were significantly different from the 1000cs and 5000cs SO tubes (P < 0.05 for each), but not the heat control tubes (P = 0.51). The mean ± SD inner lumen area expanded for tubes in 1000cs and 5000cs SO by 11.75 ± 1.7% and 2.70 ± 0.6%, respectively, but contracted for tubes in BSS by -2.70 ± 2.7% (P < 0.01 for 1000cs SO vs. BSS; P = 0.068 for 5000cs SO vs. BSS). Similarly, the mean ± SD outer lumen area expanded by 10.80 ± 0.3% and 5.24 ± 5.4% for tubes in 1000cs and 5000cs SO, respectively. Whereas tubes in BSS and heat control tubes contracted by -7.87 ± 1.0% and -1.36 ± 1.7%, respectively. When compared to the BSS at 65˚C group, the differences in mean changes of outer lumen area were statistically significant (P < 0.01 for 1000cs SO; P = 0.048 for 5000cs; P < 0.01 for heat control tubes) (Fig 3).

### Changes in flow rates through FP7 tubes after aging

Post-aging at 65˚C, the mean ± SD flow rates increased for 1000cs and 5000cs SO tubes by 61.0 ± 46.5% and 98.6 ± 58.9%, respectively. In contrast, flow rates decreased for BSS tubes by

## Dimensional changes in FP7 tubes after aging in silicone oil

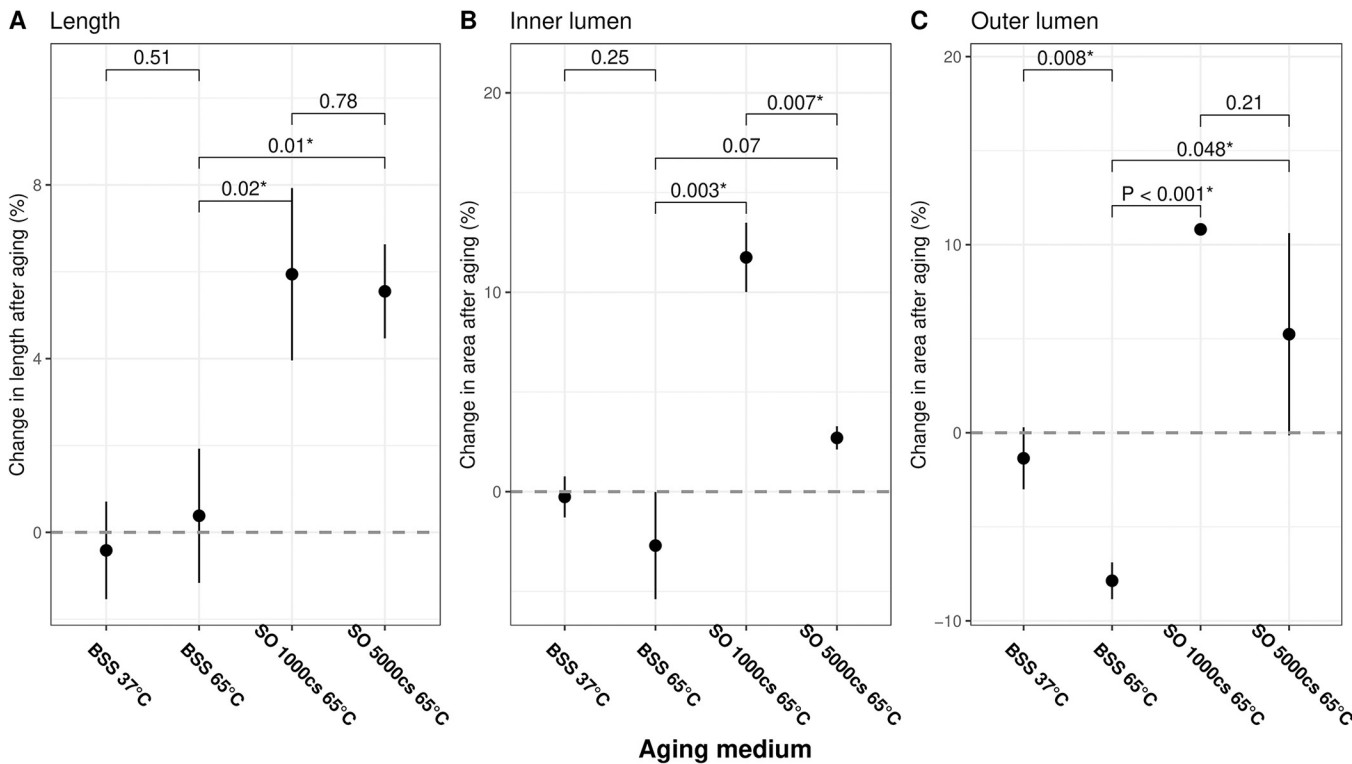

**Fig 3. Dimensional changes in FP7 tube segments after aging.** The x-axes represent each aging medium with their respective temperatures. The y-axes represent the mean % change in tubes' (A) lengths, (B) inner lumen area, and (C) outer lumen area. The central black dot represents the mean, with vertical bars representing one standard deviation. The horizontal bars indicate comparisons between two groups with P-values of the two-tailed T-tests on top. The horizontal dashed line marks a 0% change. * denotes statistical significance at < 0.05. BSS = balanced salt solution, SO = silicone oil.

-4.92 ± 8.6%, which was statistically significantly lower when compared to 1000cs and 5000cs SO tubes (P < 0.01 for both) (Fig 4).

## Discussion

In this in-vitro laboratory experiment, we investigated the impact of prolonged exposure of AGV FP7 tube segments to SO. Our results indicate that accelerated aging in both 1000cs and 5000cs SO increased the cross-sectional area and expanded the length of the tubes, albeit to different extents. In contrast, aging at the same temperature in BSS had a shrinking effect on the tubes' cross-sectional area. Our findings may be explained by the potential chemical interactions that occurred during the aging process. Given the structural similarity of silicone oil (PDMS) and silicone rubber, a highly cross-linked form of PDMS [25], one can infer that silicone oil permeated into the structural matrix of FP7 tubes, causing swelling. Hildebrand and Hansen solubility parameters—which predict the degree of dissolution of a polymer into a solvent based on their structural similarity—further support this inference [36, 37]. The viscosity of PDMS polymers depends on the polymer's molecular weight. A lower viscosity implies a lower molecular weight due to shorter polymer chains. Hence, the shorter polymer chains of the 1000cs were more mobile and could permeate the silicone elastomer matrix more efficiently than the 5000cs PDMS [38]. Thereby, our results indicate that 5000cs SO penetrated the matrix of silicone rubber to a lesser extent than its 1000cs counterpart as the percentage increase in the inner and outer lumen area was higher in tubes aged in 1000cs, with the former metric being statistically significant (Fig 3B & 3C). Although, the changes in the flow rates were not significantly different between the two SO groups (Fig 4).

SO-induced expansion of AGV tube lumens is potentially clinically significant as it can impact the rate of aqueous fluid drainage and may facilitate SO migration. Our results suggest that 1000cs SO may pose an increased risk of sub-conjunctival SO migration due to its ability to permeate through the FP7 tube matrix readily. Previous studies have suggested that this migration may occur in various ways, including through the tube itself, through poorly sutured sclerotomies, or via leakage between the tube and the sclera [39–41]. Some studies have also reported a higher rate of in vivo emulsification of 1000cs SO oil than 5000cs [19, 42, 43], which may further enhance SO migration. Future studies may emphasize the role of SO viscosity in the success rates of AGV.

Moreover, our study indicates that non-silicone-based GDDs may be a better choice for refractory SO-induced glaucoma. One such example is EX-PRESS® (Alcon Inc., Fort Worth, TX, USA) metal shunt, which avoids SO-silicone rubber interactions altogether and clinically has shown similar or higher success rates compared to AGV [44–46]. Notably, because SO can cause Glaucoma through more than one mechanism [13], more clinical studies are needed to validate this option further.

The reason behind the shrinkage of the cross-sectional area of tubes aged in BSS, specifically the outer lumen area, at 65˚C is not readily apparent. One potential explanation is that silicone rubbers possess varying heat resistance based on the degree of cross-linking, length of the polymers, and incorporation of additives [47–49]. Accordingly, the radial shrinkage of tubes aged in BSS may be attributed to further cross-linking or curing of the tubes, likely due to heat [47, 50, 51]. In contrast, tubes aged in 1000cs and 5000cs SO were heated in an expanded network polymer configuration due to the intrusion of SO within the tube matrix, which likely offered active resistance to shrinking. Tubes aged in BSS had free space for molecular mobility and thus relatively experienced more of the shrinking effect—a phenomenon commonly reported with PDMS-derived structures [51–53].

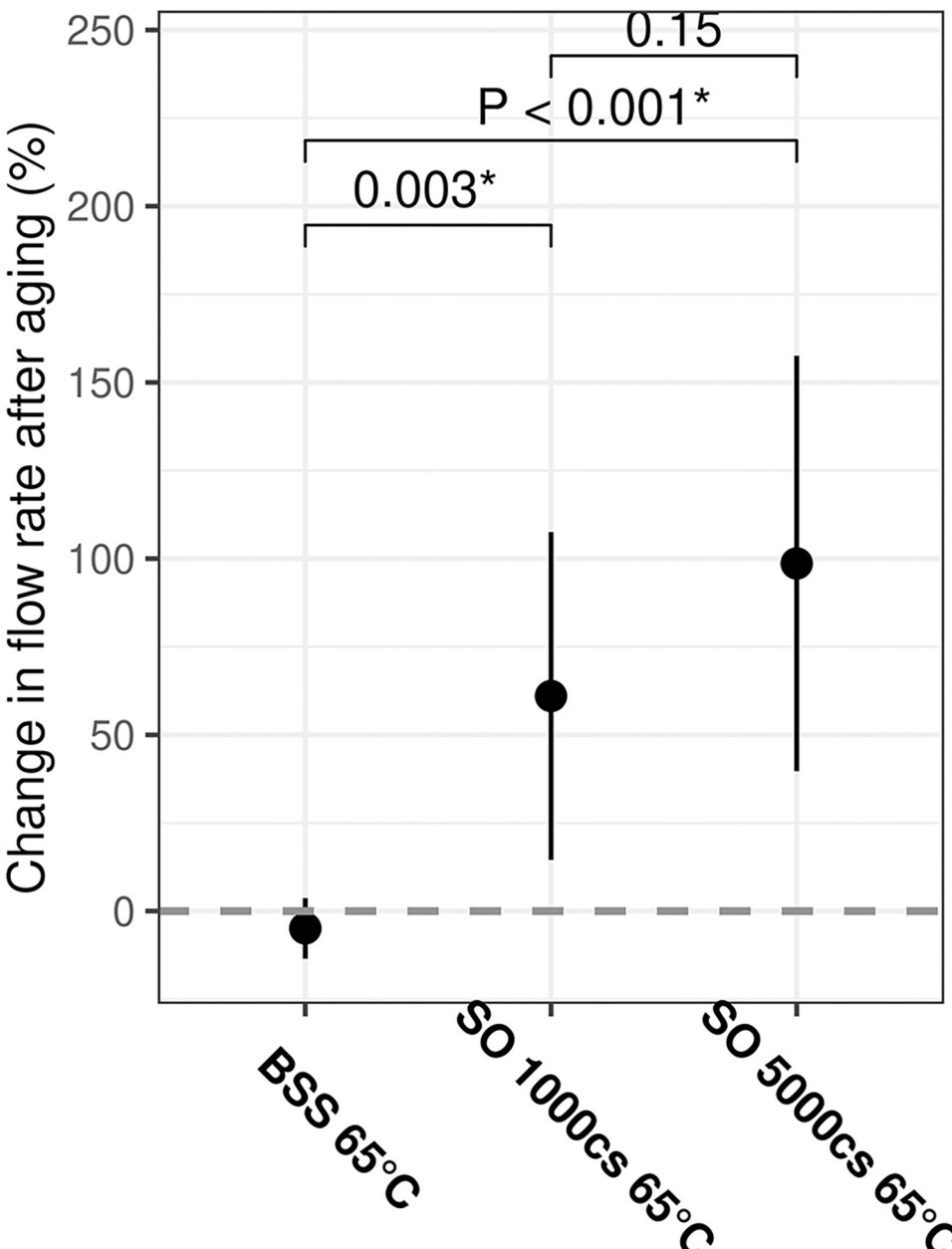

**Fig 4. Results of the gravity flow test.** The x-axes represent each aging medium with their respective temperatures. The y-axes represent the mean % change in the tubes' flow rate of BSS. The central black dot represents the mean, with vertical bars representing one standard deviation. The horizontal bars indicate comparisons between two groups with P-values of the two-tailed T-tests on top. The horizontal dashed line marks a 0% change. * denotes statistical significance at < 0.05. BSS = balanced salt solution. SO = silicone oil.

Our gravity flow test demonstrated that tubes aged in either 1000cs or 5000cs SO increased in flow rates, whereas those aged in BSS experienced minimal change. Increased flow rates in SO-aged tubes can partially be explained by their increased lumen area (Fig 3B). Tubes aged in 5000cs SO experienced a lesser degree of lumen expansion than those aged in 1000cs SO but displayed a greater degree of increase in flow rate. Albeit not statistically significant, this difference can be attributed to a greater degree of SO deposition or coating in the lumen of tubes aged in 5000cs SO.

Changes in luminal flow rates likely have a greater clinical significance in non-valved GDDs that lack any other means of proximal flow restriction. Nonetheless, while AGV has lower rates of post-operative hypotony due to a valve mechanism [2], AGV's valve is also composed of silicone and may be subjected to interactions with SO. As such, migration of SO through the tube may affect the drainage of aqueous fluid by increasing the lumen size, as observed in our experiment, and by potentially changing valve function [54]. While beyond the scope of our current work, these complex interactions would benefit from additional future investigations.

Our study has several significant limitations. In clinical practice, the drainage of aqueous flow through GDDs depends on several factors such as the location of the tube insertion, kinking of the tube [55], age of the patient, extent of fibrous encapsulation of the bleb [56], and protein content of the aqueous humor [2]–all of which can potentially confound a direct comparison of flow rates. Furthermore, the generalizability of our results is limited by the fact that we used silicone tubes only from the AGVs, which may differ in composition from silicone tubes in other GDDs.

## Conclusion

Overall, this controlled laboratory experiment showed that prolonged exposure of silicone-based AGV tubes to SO impacted the dimensions and flow characteristics of the tubes. Surprisingly, it did not impede flow through the FP7 tubes and, in fact, increased flow due to the increased diameter of the lumen. The increase in flow rates may carry clinical implications such as post-operative hypotony, especially for non-valved GDDs. Future studies may expand on our results by investigating the impact of SO exposure on different GDDs' functionality using an *in vitro* model which may offer additional insights regarding the changes in flow dynamics.

## Supporting information

**S1 Dataset. Experimental data.** The experimental data used to generate the results in the manuscript can be found in the sheets "10min%chngeflowrates" and "%change_values". (XLSX)

## Acknowledgments

The authors thank New World Medical Inc. for supplying Ahmed glaucoma valves and Forsight Laboratories for allowing us to use their research facilities.

## Author Contributions

**Conceptualization:** Abu Tahir Taha, Ying Han, Frank Brodie.

**Data curation:** Abu Tahir Taha.

**Formal analysis:** Abu Tahir Taha.

**Investigation:** Abu Tahir Taha, Ying Han.

**Methodology:** Abu Tahir Taha, Matthew Clarke, Ying Han.

**Project administration:** Abu Tahir Taha.

**Resources:** Abu Tahir Taha, Matthew Clarke, Frank Brodie.

**Software:** Abu Tahir Taha.

**Supervision:** Frank Brodie.

**Validation:** Abu Tahir Taha.

**Visualization:** Abu Tahir Taha, Chiara Wabl.

**Writing – original draft:** Abu Tahir Taha.

**Writing – review & editing:** Abu Tahir Taha, Chiara Wabl, Ying Han, Frank Brodie.

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
