## [Decision Letter · Decision Letter 0]

9 Jul 2024

PONE-D-24-14929Flow characterization of Ahmed glaucoma FP7 tubes after in-vitro aging in silicone oilPLOS ONE

Dear Dr. Brodie,

Thank you for submitting your manuscript to PLOS ONE. After careful consideration, we feel that it has merit but does not fully meet PLOS ONE’s publication criteria as it currently stands. Therefore, we invite you to submit a revised version of the manuscript that addresses the points raised during the review process.

We look forward to receiving your revised manuscript.

Kind regards,

Eleftherios Paschalis Ilios

Academic Editor

PLOS ONE

Journal Requirements:

Additional Editor Comments:

Both reviewers see importance in this work. Please address comments of Reviewer 2 in order to consider this manuscript further.

Pay attention to clarity.

Thank you

Reviewers' comments:

Reviewer's Responses to Questions

**Comments to the Author**

1. Is the manuscript technically sound, and do the data support the conclusions?

Reviewer #1: Yes

Reviewer #2: Partly

2. Has the statistical analysis been performed appropriately and rigorously? 

Reviewer #1: Yes

Reviewer #2: Yes

3. Have the authors made all data underlying the findings in their manuscript fully available?

Reviewer #1: Yes

Reviewer #2: Yes

4. Is the manuscript presented in an intelligible fashion and written in standard English?

Reviewer #1: Yes

Reviewer #2: No

5. Review Comments to the Author

Reviewer #1: The reviewer has gone through the study titled “Flow characterization of Ahmed glaucoma FP7 tubes after in-vitro aging in silicone oil” with keen interest and enthusiasm. Reviewer compliments authors /researcher for taking up this very novel concept to analyze the functional integrity of AGV tube in a highly specific clinical entity of Silicone oil-induced secondary glaucoma. The study has brought out a very interesting observation on the structural and functional changes in the silicone tube including the flow of silicon oil through the tube which is going to add to the present knowledge on the delayed changes in the AGV tube following AGV Implant surgery.

Retraction of AGV tube is a known entity. There is clinical evidence of thinning of the scleral patch or flap as well as of the conjunctiva due to delayed reaction or interaction between silicone material of AGV tube with ocular tissues. However, the literature is silent on the very important issue of intra-lumen changes and the facility of the intra-lumen flow of the aqueous through the AGV Tube. Despite the interesting observation of widening of the tube lumen and increased flow of aqueous through the tube in the case of silicone oil-induced glaucoma, the post-AGV surgery, control of Intraocular pressure remains debatable in the post Silicone induced secondary glaucoma. In the reviewer’s view, the 360-degree clogging of trabecular meshwork due to saponification of silicone Oil and excessive fibrosis of conjunctiva due to the presence of silicone oil in the subconjunctival tissue as well as around the valve plate invariably lead to the compromised functional outcome. Reviewer compliments authors for such novel study and in concurrence with them.

Reviewer #2: Please find the comments in the attached file.

6. PLOS authors have the option to publish the peer review history of their article (what does this mean?). If published, this will include your full peer review and any attached files.

Reviewer #1: **Yes: **MAJ GEN JKS PARIHAR

Reviewer #2: No

---

## [Author Response · Author response to Decision Letter 0]

1 Aug 2024

Please see attached document with comprehensive response to reviewers. Thank you

---

## [Decision Letter · Decision Letter 1]

2 Sep 2024

Flow characterization and structural alterations in Ahmed glaucoma FP7 tubes after in-vitro aging in silicone oil

PONE-D-24-14929R1

Dear Dr. Brodie,

We’re pleased to inform you that your manuscript has been judged scientifically suitable for publication and will be formally accepted for publication once it meets all outstanding technical requirements.

Kind regards,

Eleftherios Paschalis Ilios

Academic Editor

PLOS ONE

Additional Editor Comments (optional):

Thank you for revising the manuscript. 

Reviewers' comments:

Reviewer's Responses to Questions

**Comments to the Author**

1. If the authors have adequately addressed your comments raised in a previous round of review and you feel that this manuscript is now acceptable for publication, you may indicate that here to bypass the “Comments to the Author” section, enter your conflict of interest statement in the “Confidential to Editor” section, and submit your "Accept" recommendation.

Reviewer #1: All comments have been addressed

2. Is the manuscript technically sound, and do the data support the conclusions?

Reviewer #1: Yes

3. Has the statistical analysis been performed appropriately and rigorously? 

Reviewer #1: Yes

4. Have the authors made all data underlying the findings in their manuscript fully available?

Reviewer #1: Yes

5. Is the manuscript presented in an intelligible fashion and written in standard English?

Reviewer #1: No

6. Review Comments to the Author

Reviewer #1: Manuscript Number PONE-D-24-14929

"Flow characterization and structural alterations in Ahmed glaucoma FP7 tubes after in-vitro aging in silicone oil"

The reviewer has gone through the study titled “Flow characterization of Ahmed glaucoma FP7 tubes after in-vitro aging in silicone oil” with keen interest and enthusiasm. Reviewer compliments authors /researcher for taking up this very novel concept to analyze the functional integrity of AGV tube in a highly specific clinical entity of Silicone oil-induced secondary glaucoma. The study has brought out a very interesting observation on the structural and functional changes in the silicone tube including the flow of silicon oil through the tube which is going to add to the present knowledge on the delayed changes in the AGV tube following AGV Implant surgery.

Silicone oil (SO) is widely used in the management of complicated retinal detachment. However, intraocular SO tamponade can cause a series of complications including transient or permanent intraocular pressure (IOP) elevation1. A major reason for the occurrence of glaucoma secondary to SO tamponade is SO emulsification. The underlying mechanism might be the migration of SO droplets into the anterior chamber, which could directly obstruct the trabecular meshwork by inflammation or fibrosis2.Retraction of AGV tube is a known entity. There is clinical evidence of thinning of the scleral patch or flap as well as of the conjunctiva due to delayed reaction or interaction between silicone material of AGV tube with ocular tissues.

However, the literature is silent on the very important issue of intra-lumen changes and the facility of the intra-lumen flow of the aqueous through the AGV Tube. Despite the interesting observation of widening of the tube lumen and increased flow of aqueous through the tube in the case of silicone oil-induced glaucoma, the post-AGV surgery, control of Intraocular pressure remains debatable in the post Silicone induced secondary glaucoma. The article contains very high scientific content as well as brought out a very interesting observation of widening of the lumen and elongation of the length of the AGV tube which opposite to the pattern of post operative changes in the AGV Tube.

In the reviewer’s view, the 360-degree clogging of trabecular meshwork may be due to saponification of silicone Oil and excessive fibrosis of conjunctiva due to the presence of silicone oil in the subconjunctival tissue. A laboratory analysis of intraocular fluid and vitreous specimens obtained from patients undergoing removal of silicone oil showed the presence of transforming growth factor beta (TGF-β2), basic fibroblast growth factor (bFGF), interleukin 6 (IL-6), all of which can lead to fibrosis of the tube and alter the flow characteristics3. Also, all eyes had undergone PPV which led to the formation of subconjunctival scarring4.

The present study states that due to prolonged exposure to SO, the AGV FP7 was structurally altered by expanding in cross sectional area. The lesser viscosity 1000cs SO was able to permeate the silicon AGV FP7 tube better than the higher viscosity 5000cs SO. But, due to its increased mobility, the 1000cs SO is more likely to block the TM and cause higher IOP elevation5. This is in addition to the other potential effects of altering the effects of aqueous fluid drainage due to the widening of tube lumen.

In conclusion, as mentioned in the study that a laboratory-controlled experiment might not be as precise to study the drainage of aqueous flow through a GDD as in clinical practice, due to a variety of confounding factors which cannot be generalised for each patient. But future studies may expand on the present findings and include in-vitro models along with GDDs of other types using different varieties of silicon.

Reviewer compliments authors for such novel study and in concurrence with them.

REFERENCES

1. 1 Lucke K. H., Foerster M. H., and Laqua H., Long-term results of vitrectomy and silicone oil in 500 cases of complicated retinal detachments, American Journal of Ophthalmology. (1987) 104, https://doi.org/10.1016/0002-9394(87)90176-0, 2-s2.0-0023546675

2. Qian Z, Xu K, Kong X, Xu H. Ahmed Glaucoma Valves versus EX-PRESS Devices in Glaucoma Secondary to Silicone Oil Emulsification. J Ophthalmol. 2018;2018: 8539689. 2 doi:10.1155/2018/8539689

3. Asaria RHY, Kon CH, Bunce C, Sethi CS, Limb GA, Khaw PT, et al. Silicone oil concentrates fibrogenic growth factors in the retro-oil fluid. Br J Ophthalmol. 2004;88: 1439–1442. doi:10.1136/bjo.2003.040402

4. Nguyen Q. H., Lloyd M. A., Heuer D. K. et al., Incidence and management of glaucoma after intravitreal silicone oil injection for complicated retinal detachments, Ophthalmology.

5. Petersen J. and Ritzau-Tondrow U., Chronic glaucoma following silicone oil implantation: a comparison of two oils of differing viscosity, Fortschritte der Ophthalmologie. (1988) 85, no. 6, 632–634, in German.

7. PLOS authors have the option to publish the peer review history of their article (what does this mean?). If published, this will include your full peer review and any attached files.

Reviewer #1: **Yes: **MAJ GEN (prof)JKS PARIHAR ,RETIRED

---

## [Editor Report · Acceptance letter]

5 Sep 2024

PONE-D-24-14929R1 

PLOS ONE

Dear Dr. Brodie, 

I'm pleased to inform you that your manuscript has been deemed suitable for publication in PLOS ONE. Congratulations! Your manuscript is now being handed over to our production team.

Kind regards, 

on behalf of

Dr. Eleftherios Paschalis Ilios 

Academic Editor

PLOS ONE